# A Wearable Sandwich Heterostructure Multimode Fiber Optic Microbend Sensor for Vital Signal Monitoring

**DOI:** 10.3390/s24072209

**Published:** 2024-03-29

**Authors:** Fumin Zhou, Binbin Luo, Xue Zou, Chaoke Zou, Decao Wu, Zhijun Wang, Yunfang Bai, Mingfu Zhao

**Affiliations:** 1Chongqing Key Laboratory of Optical Fiber Sensor and Photoelectric Detection, Chongqing University of Technology, Chongqing 400054, China; zhoufumin@stu.cqut.edu.cn (F.Z.); 15703079168@163.com (C.Z.); wudecao@163.com (D.W.); wangzhijun@stu.cqut.edu.cn (Z.W.); 5321713101@stu.cqut.edu.cn (Y.B.); zmf@cqut.edu.cn (M.Z.); 2School of Communications and Information Engineering, Chongqing University of Posts and Telecommunications, Chongqing 400065, China

**Keywords:** sandwich heterostructure multimode optical fiber, microbend sensor, heart rate, respiratory rate, ballistocardiography, wearable devices

## Abstract

This work proposes a highly sensitive sandwich heterostructure multimode optical fiber microbend sensor for heart rate (HR), respiratory rate (RR), and ballistocardiography (BCG) monitoring, which is fabricated by combining a sandwich heterostructure multimode fiber Mach–Zehnder interferometer (SHMF-MZI) with a microbend deformer. The parameters of the SHMF-MZI sensor and the microbend deformer were analyzed and optimized in detail, and then the new encapsulated method of the wearable device was put forward. The proposed wearable sensor could greatly enhance the response to the HR signal. The performances for HR, RR, and BCG monitoring were as good as those of the medically approved commercial monitors. The sensor has the advantages of high sensitivity, easy fabrication, and good stability, providing the potential for application in the field of daily supervision and health monitoring.

## 1. Introduction

With economic development and social progress, public health awareness is gradually increasing. The continuous online monitoring of vital signs (e.g., blood pressure, heart rate (HR), heartbeat, respiratory rate (RR), blood oxygen saturation, blood sugar, etc.) is becoming more and more important [1], which can help evaluate and diagnose specific diseases at an early stage [2,3]. Among these vital signs, RR and HR are the most common indicators that are related to a wide range of age-related diseases that are closely associated with a variety of geriatric diseases, such as cardiac arrhythmias and coronary artery disease [4].

In the past, many sensors for measuring human RR and HR have been proposed, including piezoelectric-based sensors [5], electromagnetic wave scanning [6], fiber optic sensors [7,8,9,10,11], etc. Among them, fiber optic sensors have attracted the attention of many researchers due to the advantages of high sensitivity, anti-electromagnetic interference, and lightweight. The optical fiber sensor primarily achieves HR monitoring by detecting both the heartbeat signal and the ballistocardiography (BCG) signal. The key distinction between the two lies in the fact that the BCG signal exhibits a greater number of characteristic peaks within a single heartbeat cycle, accompanied by a relatively low signal intensity. Most of the usually used schemes was to embed optical fiber sensors into a mattress to monitor the RR and HR of a human lying on the bed [7,8,9,10,11,12,13,14,15,16,17,18]; however, it would require a relatively large sensing area, thus leading to the sensing optical fiber being too long. Another way was to embed the designed optical fiber sensors in a belt to construct a wearable device [19] that was capable of monitoring RR and HR even when the subject was standing or moving. In 2022, Tavares et al. [20] reported a wearable FBG sensor that was encapsulated in 3D printable materials for RR and HR monitoring, but it required expensive wavelength modulation equipment. In 2023, Zha et al. [21] developed a wearable belt embedded with an elastomer optical fiber section between two multimode silica optical fibers for RR and HR monitoring, exhibiting good accuracy and stability. In 2024, Wang et al. [22] proposed a novel wearable optical microfiber intelligent sensor based on wave-shaped polymer optical microfibers for the monitoring of human cardiopulmonary functions and behaviors. This sensor boasts the advantages of high sensitivity and potential for stability.

Nevertheless, since the HR signal is induced from human heartbeat and conducted to the outer left thoracic surface, the amplitude of the HR signal is usually much weaker than that of the RR signal that is induced by the contraction of the external chest cavity. In order to precisely extract the HR signal from the time domain mixed signal of HR and RR, it is also expected that the wearable sensor has a similar sensitivity to both of them. Since most of the RR and HR optical fiber sensors are based on the principle of sensing the pressure or bending induced by respiration and heartbeat, microbending is usually utilized to increase the pressure or strain sensitivity of the optical fiber [23,24], including the mattress type [14,15,16,25,26] and the wearable belt type [19].

In this work, we propose a wearable optical fiber microbend sensor constructed by combining a sandwich heterostructure multimode fiber Mach–Zehnder interferometer (SHMF-MZI) with a microbend deformer for the simultaneous measurement of RR and HR, which could greatly enhance the response for the HR signal. The basic concept and structure of the SHMF-MZI have been proposed in our previous work [27], which could achieve extremely high curvature sensitivity in a certain range. Herein, the structure parameters of the SHMF-MZI are optimized to adapt to the application of RR and HR measurement, and a microbend deformer is designed to enhance the pressure sensitivity induced by the microbending loss and overcome the instability during the measurement. The BCG signal is also successfully extracted from the detected waveforms. The proposed wearable sensor has the advantages of very high sensitivity, simple fabrication, and good stability, and it indirectly contacts human skin, thus enabling the comfortable and continuous monitoring of vital signs.

## 2. Methodology

### 2.1. Principle of SHMF-MZI

In this study, the single mode fiber (SMF), step-index multimode fiber (SIMMF) and graded-index multimode fiber (GIMMF) employed in the creation of the SHMF-MZI were sourced from Yangtze Optical Fiber Company. The structure of the SHMF-MZI is illustrated in Figure 1, which is constructed using a section of GIMMF spliced with two sections of SIMMF on both of its ends, and the diameter of the GIMMF is smaller than that of the SIMMF. When light transmits from the input-SMF into SIMMF1, due to the mismatch between the fiber core diameters, higher-order core modes are stimulated in SIMMF1. When light transmits to the GIMMF, due to the mismatching of the mode fields between GIMMF and SIMMF1, part of the light is coupled to the core modes of the GIMMF, and the rest is coupled to the cladding modes of the GIMMF and finally recoupled into the high-order core modes of SIMMF2. Light in the core of the GIMMF is propagated along the sinusoidal curve-like trajectory and also finally recoupled into the core of SIMMF2, thus finally forming an in-line optical fiber interferometer called SHMF-MZI. The specially designed structure of the SHMF-MZI could achieve extremely high curvature sensitivity within a small range of bends [27].

The transmission of the SHMF-MZI sensor can be given by the following:(1)$$I=I_{\mathrm{core}}+I_{\mathrm{clad}}+2\sqrt{I_{\mathrm{core}}I_{\mathrm{clad}}}\cos\frac{2\pi L(n_{\mathrm{core}}-n_{\mathrm{clad}})}{\lambda }$$
where I is the output light intensity, $I_{\mathrm{core}}$ and $I_{\mathrm{clad}}$ are the light intensity of the core and cladding modes of the GIMMF, respectively, $n_{\mathrm{core}}$ and $n_{\mathrm{clad}}$ are the effective refractive indices of the core and cladding of the GIMMF, respectively, L is the length of the GIMMF, and λ is the wavelength.

The free spectral range (*FSR*) of the SHMF-MZI sensor is given by the following:(2)$$FSR=\frac{\lambda^{2}}{\Delta n\cdot L}$$
where ∆*n* is the effective refractive index difference between the core mode and cladding mode of the GIMMF. The normalized extinction ratio of the SHMF-MZI sensor can be expressed as follows:(3)$$ER=\frac{2\sqrt{I_{\mathrm{core}}I_{\mathrm{clad}}}}{I_{\mathrm{core}}+I_{\mathrm{clad}}}$$

### 2.2. Simulation and Optimization of SHMF-MZI

In order to determine the structural parameters of the SHMF-MZI sensor to achieve a desirable transmission spectrum that is suitable for the application of RR and HR measurement, the simulation is carried out using the Beam Propagation Method (BPM) in Rsoft 2018 software. The effect of the length of the SIMMF and GIMMF on the transmission spectrum of the SHMF-MZI is investigated separately using the control variable approach. The structural parameters of the optical fiber used in this work are shown in Table 1.

In the simulation, the length of the GIMMF is fixed at 2 cm, and the length of the SIMMF is set to be 0, 0.5, 1, 1.5, 2, and 2.5 mm, respectively, to compute the spectra of the SHMF-MZI in the wavelength range from 1250 nm to 1650 nm. The results are shown in Figure 2a. It can be seen that the length of the SIMMF has strong influences on the transmission spectrum. When the length of the SIMMF is 0 mm, that is, SMF is directly fused to the GIMMF, the interference spectrum does not appear, and the transmission spectrum is close to a straight line. It is because the SIMMF plays a decisive role, which enables the GIMMF to stimulate the higher-order transmission modes. On the contrary, when the length of the SIMMF exceeds 0 mm and gradually increases, the FSR of the transmission spectrum tends to decrease, leading the peaks of the spectrum to become more and more dense. Considering the overall uniformity of the transmission spectra as well as the compactness of the sensor as a whole, the length of the SIMMF is designed as 2 mm.

Subsequently, the length of the SIMMF is fixed at 2 mm, and the characteristics of the transmission spectrum are simulated under the condition that the length of the GIMMF is 1 cm, 2 cm, 5 cm, 10 cm, 20 cm, and 50 cm, respectively, and the results are shown in Figure 2b. It can be seen that the FSR of the transmission spectrum becomes smaller with the increase in the length of the GIMMF, and when the length of the GIMMF exceeds 2 cm, the spectrum becomes more and more dense, and the visibility of the stripes decreases obviously. Therefore, based on the above simulation results, the lengths of SIMMF and GIMMF for the SHMF-MZI sensor are chosen to be 2 mm and 2 cm, respectively, for the subsequent fabrication and experiments.

### 2.3. SHMF-MZI-Based Microbend Deformer Sensor

The microbend deformer can make the optical fiber bend regularly to achieve intensity demodulation, which is also suitable for the SHMF-MZI sensor. In the microbend deformer, the SHMF-MZI sensor experiences multi-segment bending as a result of applied forces, which effectively enhances the pressure sensitivity of the sensor. The transmission coefficient of light in the SHMF-MZI sensor is the function of pressure ∆F; by extracting the intensity change in the output light, the force information received by the sensor can be demodulated. A typical microbend deformer transducer is sawtooth-shaped with two main parameters, including microbend spacing Λ and the number of microbend spacings *N*. For GIMMF, there is an optimal microbend spacing $\Lambda_{C}$ for the microbend deformer given by [18] the following:(4)$$\Lambda_{C}=\frac{2\pi \mathrm{an}}{N.A.}$$
where *a* is the radius of the fiber core, *n* is the refractive index of the fiber core, and *N.A.* is the numerical aperture of the fiber. When Λ satisfies Equation (4), the transmission mode of the SHMF-MZI sensor reaches full coupling, and the microbend loss and sensitivity reach the maximum. Moreover, the sensitivity of the fiber optic microbend sensor increases with the number of microbend *N* [25,26]. Therefore, it is anticipated that the sensing region of the fiber optic sensor, which is the GIMMF, is fully covered by the microbend deformer.

The schematic diagram of the SHMF-MZI-based microbend sensor is shown in Figure 3a. The microbend deformer was fabricated by many sawtooth units pasted on the cover material with strong glue, and each unit was produced by injecting the silicone tube with UV glue, which was then cured to a solid state under a UV light. The optimal microbend spacing is 1.11 mm as calculated by Equation (4). For the purpose of fabrication accuracy, we set the diameter of the silicone tube to be 1 mm; that is, the microbend spacing Λ was 1 mm. Since the length of the GIMMF was designed as 2 cm, the number of microbend *N* was calculated to be 18. Considering that the microbend spacing changes with the change in elastic material for most of the knitted microbend deformer [19], in this work, the silicone tubes were attached to the back side of an inelastic Velcro with a strong adhesive, and then the SHMF-MZI sensor was attached directly to the belt using the inelastic Velcro, which made the belt easy to clean and replace. In addition, a piece of inelastic material was sewn into the belt, which would ensure that the belt could be freely stretched without affecting the sensing area. The top view and side view of the SHMF-MZI-based microbend deformer sensor are shown in Figure 3b,c, respectively.

It has been proven previously that the interference intensity of the SHMF-MZI sensor is nearly independent of the temperature [27]. The microbend deformer used in this work has a very low thermal conductivity coefficient. Consequently, the influence of temperature on the SHMF-MZI microsensor can be neglected. In order to analyze the response of the SHMF-MZI-based microbend sensor to micro-pressure, experiments were conducted by adding different standard micro-pressures to the sensor. As shown in Figure 4, light emitted from a broadband light source (BBS, CONQUER ASE 1528 nm–1610 nm) passes through the SHMF-MZI microbend sensor, and the spectra are recorded by an optical spectral analyzer (OSA, AQ6370D, 600–1700 nm). Weights of 0–7 g corresponding to pressures of 0–0.0686 N, with a step size of 1 g, were placed on the SHMF-MZI sensor in turn.

Figure 5a shows the transmission spectrum of the SHMF-MZI sensor before and after encapsulation with the microbend deformer. It can be seen that the transmission spectrum of the SHMF-MZI sensor after encapsulation is better than that before encapsulation. The FSR of the interference spectrum is nearly unchanged while the visibility of the interference fringes is significantly improved, which can be attributed to the fact that the microbend deformer has improved the coupling efficiency of the SHMF-MZI. Figure 5b,c show the transmission spectrum and the corresponding intensity near a wavelength of 1590 nm under different pressures, respectively, indicating that the intensity at around 1590 nm almost linearly varies with the pressure and the pressure sensitivity is ~98 dB/N, which is about 327, 778, and 34 times higher than that of FBGs [28], plastic optical fibers [29], and thermoplastic polyurethane optical fiber [30], respectively. Therefore, the proposed SHMF-MZI-based microbend sensor is highly sensitive to micro-pressure and very suitable for detecting vital signal parameters such as HR, RR, BCG, etc.

## 3. Experiment and Discussion

### 3.1. HR and RR Monitoring Experiments

The schematic diagram of the HR and RR monitoring system is shown in Figure 6. The wavelength of the tunable laser (TLS, wavelength range of 1528~1610 nm) was set at 1594 nm, which was within the linear working region, as shown in Figure 5a. The fiber optic isolator was used to block the backscattering and reflecting light. The wearable belt was worn on the subject’s chest, and the SHMF-MZI microbend sensor was positioned within the impulsive region of the apical impulse. The apical impulse is typically localized in the fifth intercostal space along the left midclavicular line, enabling a more effective detection of heart contractions. At the same time, when the subject is female, it can better avoid the inconvenience caused by the physiological structure. When the subject breathed and their heart beat, the microbend deformer would repeatedly apply force on top of the sensor, thus inducing intensity variations for the transmitting light, which was detected by the photodetector (PD) and then converted into voltage changes. Finally, the respiratory, heartbeat, and BCG signals could be extracted from the recorded wave.

In order to evaluate the accuracy of the wearable SHMF-MZI microbend sensor, a commercial device called Oxygen Saturation (SpO2) sensor was selected for the simultaneous measurement of HR [7], and the RR measurement results were compared with the manual counting method. The respiratory and heartbeat signals of 10 volunteers (5 males and 5 females) were collected in the experiments; all of them did not have any cardiac diseases or take any hormonal drugs prior to the test. The information about the volunteers is shown in Table 2. In the experiments, the HR and RR were measured in three postures with normal conditions: sitting, standing, and lying. During the measurement, the reference HR signal was measured by the SpO2 sensor fixed on the volunteer’s finger, and the number of rise and fall instances of the volunteer’s chest was recorded as the reference for the respiratory signal. The repeatability and stability of the SHMF-MZI microbend sensor were thoroughly evaluated through a well-designed experimental program. This included meticulous daily measurements on two volunteers, conducted over a period of five consecutive days.

Figure 7a shows the raw signal of 10 s for one of the subjects. It can be seen that the raw signal is the superposition of the respiratory signal and the heartbeat signal. To obtain the pure respiratory signal, it can be extracted by using a low-pass FFT filter with a cutoff frequency of 0.6 Hz [31], as shown in Figure 7b. When dealing with non-stationary signals, the FFT filter tends to induce phase drift, a phenomenon that can lead to considerable inaccuracies in the detection and extraction of heartbeat signals. In contrast, zero-phase bandpass filters are adept at circumventing this issue, ensuring more precise processing throughout the signal analysis. Since the effective components of the heartbeat signal with an HR of 50–110 bpm were mainly concentrated in the range of 0.8–1.8 Hz, a zero-phase bandpass filter with a frequency of 0.8–1.8 Hz was used for the recovery, as shown in Figure 7c. More importantly, different from the other type of RR and HR optical fiber sensors [7,12,19,21], owing to the combined advantages of the high bending sensitivity of the SHMF-MZI and the optimized designation of the microbending deformer, the amplitude of the extracted HR signal was comparable with that of the extracted RR.

Based on the respiration and heartbeat waveforms shown in Figure 7b,c and by using the peak tracking method, HR and RR can be calculated as follows:(5)$$\mathrm{HR}=\frac{m-n}{T_{m}-T_{n}}\times 60$$
where *T_m_* and *T_n_* are the time used for the *m*th and *n*th feature peaks, respectively.

### 3.2. Error Analysis and Discussion

The accuracy of HR and RR measured by the SHMF-MZI microbend sensor and reference sensor was analyzed by using mean error (ME), mean absolute error (MAE) and mean absolute percentage error (MAPE), which are expressed as follows [20]:(6)$$\mathrm{ME}=\frac{1}{n}\sum_{i=1}^{n}y_{i}-{\hat{y}}_{i}$$
(7)$$\mathrm{MAE}=\frac{1}{n}\sum_{i=1}^{n}\left|y_{i}-{\hat{y}}_{i}\right|$$
(8)$$\mathrm{MAPE}=\frac{100\%}{n}\sum_{i=1}^{n}\left|\frac{y_{i}-{\hat{y}}_{i}}{y_{i}}\right|$$
where ${\hat{y}}_{i}$ and $y_{i}$ refer to measured values and reference values for HR and RR, respectively, and n indicates the number of measurements.

In the experiment, about 20 sets of data were collected for each person in each posture, resulting in a total of approximately 600 sets. Firstly, these data were classified based on different individuals. The calculated parameters, including ME, MAE, MAPE, and the corresponding standard deviation (SD), were then analyzed to evaluate the measurement accuracy of the SHMF-MZI microbend sensor for different individuals. Similarly, these data were also categorized according to different postures, resulting in approximately 200 sets of data for each posture. Then, the calculated parameters were also used to assess the accuracy of the sensor in different postures. Finally, we calculated the 600 summarized sets of data for the SHMF-MZI microbend sensor. The above results are shown in Table 3.

The maximum value of ME for HR and RR was 2.3 bpm and 1.1 bpm, respectively. When considering the effect of absolute value, the maximum value of MAE for HR and RR was 4.9 bpm and 2.1 bpm, respectively. The maximum value of MAPE for HR and RR was 6.9% and 15%, respectively. According to the ANSI/AAMI EC13-2002 standard [32], the error of HR should be less than ±10% or ±5 bpm [33]. The above results indicated that the HRs measured by the SHMF-MZI microbend sensor were in good agreement with those measured by the SpO2 sensor for different individuals or different postures. The maximum SDs of the ME group and the MAE group were 6.3 bpm and 4.4 bpm, respectively, which was also acceptable [17]. Although the SD of the ME group was relatively large, we mainly focused more on the difference between the actual value and the reference value, so the SD of MAE was more in line with our expectations. The SD of the MAPE group also has a relatively small maximum value of 6.6%. Owing to the anatomical distinctions between females and males, including the presence of larger breasts and a relatively smaller heart with more delicate blood vessels in females, these disparities can influence the palpable strength of the heartbeat at the apical impulse. As a result, the accuracy of the HR measurements in females may be subject to a wider range of errors. The HR while lying down is generally lower than that when in a sitting or standing position, which may result in greater values for MAPE and SD.

As for the RR, MAPE exhibited that several sets of values were above 10%, which was relatively larger than that of the HR due to the fact that the normal RR in humans is generally smaller than the HR. However, the accuracy of RR was good in terms of the ME and MAE indicators. The maximum SDs of the ME group and the MAE group were 2.7 bpm and 1.8 bpm, respectively, which means that there are fewer groups with large absolute errors compared to the HR. Therefore, the accuracy of the HR and RR measured by the SHMF-MZI microbend sensor is fully consistent with this standard.

We summarized and computed all the data of HR and then compared them with the results of other comparable wearable devices, as shown in Table 4, where the commercial Apple Watch is based on Photoplethysmographic (PPG) sensors. It can be seen that the SHMF-MZI microbend sensor has the smallest ME (−0.1 bpm), while the MEs of the FBG-type sensor and Apple Watch are 0.8 bpm and −1.7 bpm, respectively. For the indicator MAE, the SHMF-MZI microbend sensor is comparable with the FBG but smaller than that of the Apple Watch. For the indicator MAPE, the SHMF-MZI microbend sensor is larger than the FBG but smaller than the Apple Watch. The SD values of all three groups are smaller than those of the reference sensors, which means that the SHMF-MZI sensor has greater accuracy. Therefore, the overall performance of the proposed sensor is satisfactory. Furthermore, the SHMF-MZI microbend sensor is based on intensity modulation, which will reduce the cost and complexity of the demodulation system.

### 3.3. BCG Signal Measurement

BCG is a micro-vibratory signal caused by the activity of the human heart, and it is a non-invasive signal that can be collected by sensors [35]. It is possible to analyze the health of the cardiovascular system and synchronize information about cardiac hemodynamics through the BCG signal. The BCG signal is weaker than the heartbeat signal and contains multiple characteristic peaks and troughs within a single heartbeat cycle, which requires very high sensitivity from the sensors. The BCG signal contains the F–N peak and the J peak, which are stronger than the other peaks, and these are associated with the entire course of the heartbeat. The J peak occurs when the heart valves are open and a large amount of blood is pumped out of the ventricles. Other characteristic peaks have their own unique significance, but HR is usually calculated using the peak tracking of the J peaks over different cycles. The SHMF-MZI microbend sensor was used to measure BCG for male volunteer 3 in a lying position. The BCG signal was extracted by a fourth-order Butterworth bandpass filter with a frequency band of 5–12 Hz.

The extracted BCG signal of the proposed sensor and the signal measured by SpO2 are demonstrated in Figure 8a. It can be seen that the J peak of the BCG signal coincided with the peak of the SpO2 signal, proving that the SHMF-MZI microbend sensor was also effective for the measurement of BCG. In all heartbeat cycles, characteristic peaks and valleys I, J, and K can be clearly seen, while the other characteristic peaks and valleys are relatively weak, which is acceptable because the calculation of heart rate mainly uses the J peak. In Figure 8b, the BCG signal is zoomed in order to show the main features, and the points from F–N features can be seen. The visibility and intensity of the J peak are the best, which is consistent with expectations. The HR can be calculated by labeling the J peak by the peak tracking method. Table 5 shows the comparison between the HR measured by BCG and SpO2 signals, indicating that the maximum absolute error is 3 bpm, which is better than that mentioned in [21]. The ME, MAE, and MAPE of the HR were also evaluated, which were 0.55 bpm, 0.85 bpm, and 1.28%, respectively, and the corresponding SDs were 1.07 bpm, 0.85 bpm, and 1.27 bpm. These values are acceptable according to the ANSI/AAMI EC13-2002 standard [32] and better than the parameters of the reference sensor in Table 4. Therefore, the SHMF-MZI microbend sensor has exhibited good function in accurately measuring BCG signal as well.

### 3.4. Correlation Analysis

The Pearson correlation coefficient (*r_X_*_,*Y*_) was also used to evaluate the linear correlation of the HR between the measurements of the SHMF-MZI microbend sensor and the SpO2 sensor and that of RR between the measurements of the SHMF-MZI microbend sensor and manual counting. The *r_X_*_,*Y*_ is defined as the following:(9)$$r_{X,Y}=\frac{\mathrm{Cov}(X,Y)}{\sigma_{X}\sigma_{Y}}$$
where *X* and *Y* are the values measured by the SHMF-MZI microbend sensor and the reference sensor, respectively. *Cov*(*X*,*Y*) is the covariance, and $\sigma_{X}$ and $\sigma_{Y}$ are the standard deviation of *X* and *Y*, respectively.

The data were categorized and then calculated for different people and different postures, as described above and as shown in Table 6. It can be seen that the maximum and minimum values of Pearson coefficients for the HR are 0.94 and 0.62, respectively, and those for the RR are 0.91 and 0.64, respectively. Generally speaking, when the Pearson coefficients of two independent measured values are greater than 0.6, this can prove that they are strongly correlated [36]. It can be seen in Table 6 that all of the Pearson coefficients are greater than 0.6 whether it is a different person or a different pose, indicating that the signal measured by the SHMF-MZI microbend sensor is in good agreement with those of the reference signals. To analyze the generalizability, Pearson coefficients for the HR and RR calculated after summarizing the data from 600 sets are 0.92 and 0.90, respectively, which can be regarded as the averaged values. Therefore, it can be concluded that the performance of the SHMF-MZI microbend sensor for HR and RR monitoring is as good as that of the medically approved commercial monitors.

## 4. Conclusions

In this study, we have designed a wearable SHMF-MZI microbend sensor for HR, RR, and BCG monitoring, which was fabricated by combining an SHMF-MZI sensor with a microbend deformer. The parameters of the SHMF-MZI sensor and the microbend deformer were analyzed and optimized in detail. The SHMF-MZI sensor was made by sandwiching a 2 cm GIMMF between two 2 mm SIMMFs. The specially designed microbend deformer could further enhance the pressure sensitivity of the SHMF-MZI, and the proposed encapsulated method could overcome the instability during the measurement. The pressure sensitivity of the SHMF-MZI microbend sensor achieved about 98 dB/N in the range of 0–0.0686 N, and more importantly, the amplitude of the extracted HR signal was comparable with that of the RR. Experimental results showed that the accuracy of HR and RR measured by the SHMF-MZI microbend sensor was fully consistent with the ANSI/AAMI EC13-2002 standard. The SHMF-MZI microbend sensor offers precise measurements. However, its accuracy in HR readings is found to be more pronounced in males than in females. The proposed wearable sensor was also used for monitoring the BCG signal, demonstrating that the maximum absolute error value was only 3 bpm. The proposed wearable optical fiber sensor has the advantages of high sensitivity, easy fabrication, very low thermal crosstalk, low price, and anti-electromagnetic interference, providing the potential for application in the field of vital supervision and health monitoring.

## Figures and Tables

**Figure 1 sensors-24-02209-f001:**
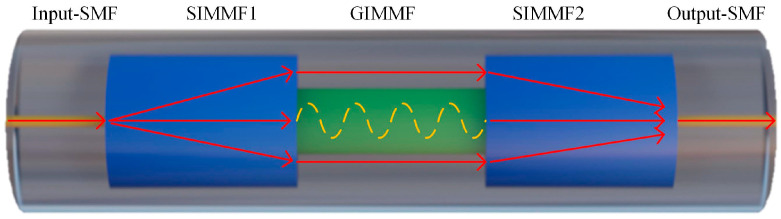
Schematic diagram of the SHMF-MZI structure.

**Figure 2 sensors-24-02209-f002:**
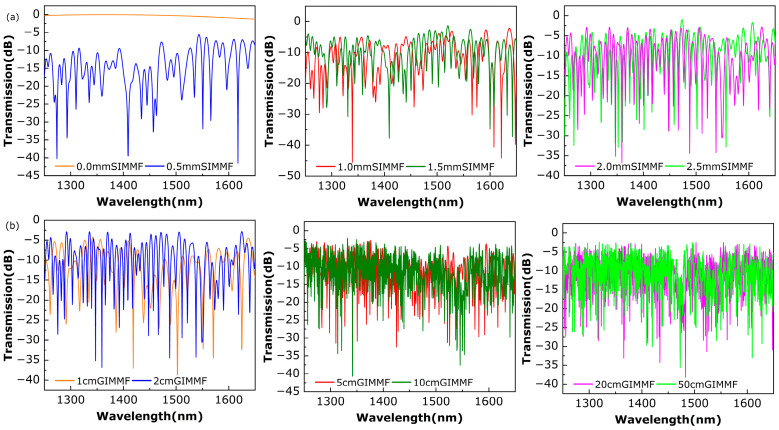
Simulation results of the effect of (**a**) SIMMF length on the transmission spectrum when the GIMMF length is 2 cm; (**b**) GIMMF length on the transmission spectrum when the SIMMF length is 2 mm.

**Figure 3 sensors-24-02209-f003:**
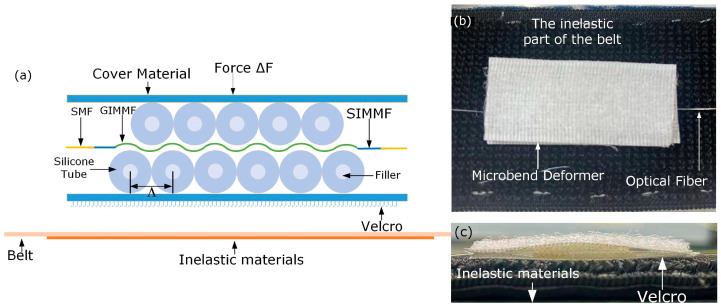
(**a**) Schematic of the SHMF-MZI encapsulated by a microbend deformer; (**b**) top view and (**c**) side view of the SHMF-MZI microbend sensor integrated with the belt.

**Figure 4 sensors-24-02209-f004:**
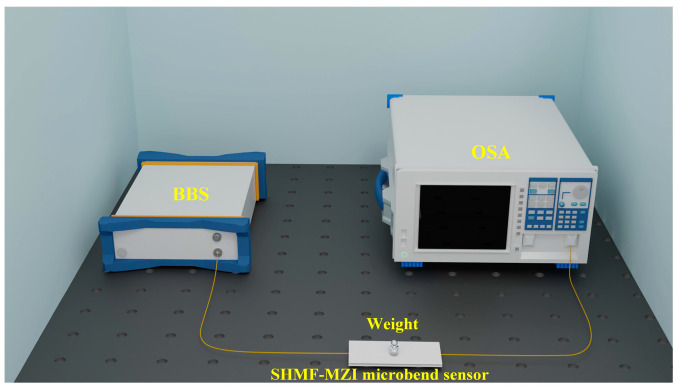
Experimental setup for the micro-pressure experimental device.

**Figure 5 sensors-24-02209-f005:**
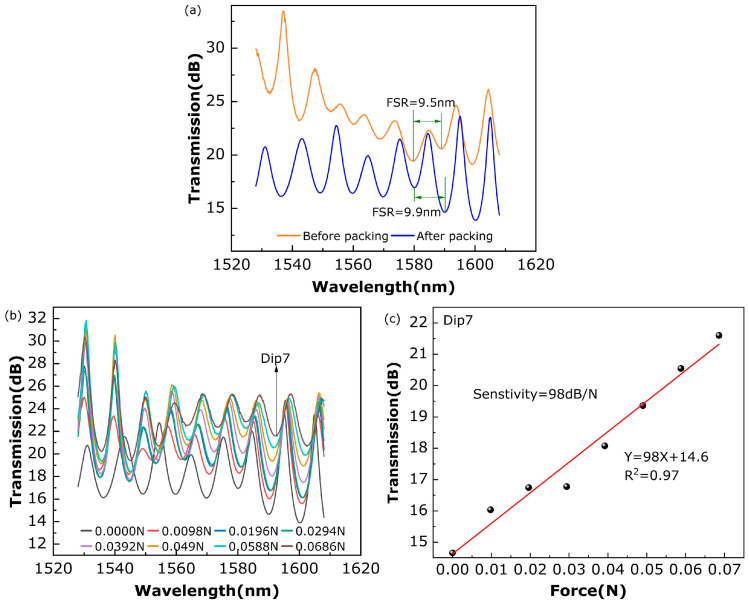
(**a**) Transmission spectra before and after fiber encapsulation, (**b**) spectrum evolution of the SHMF-MZI microbend sensor under different pressures, and (**c**) the corresponding intensity variation near a wavelength of 1590 nm.

**Figure 6 sensors-24-02209-f006:**
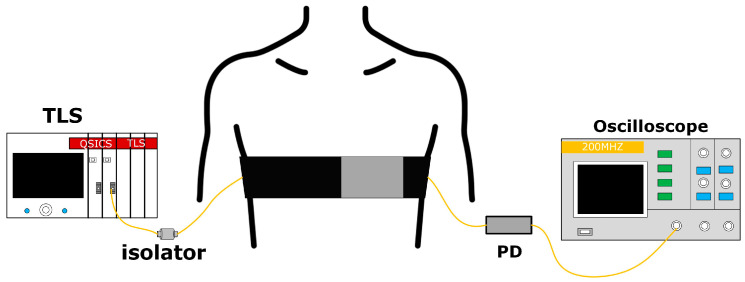
Monitoring scheme for HR and RR based on the wearable SHMF-MZI microbend sensor.

**Figure 7 sensors-24-02209-f007:**
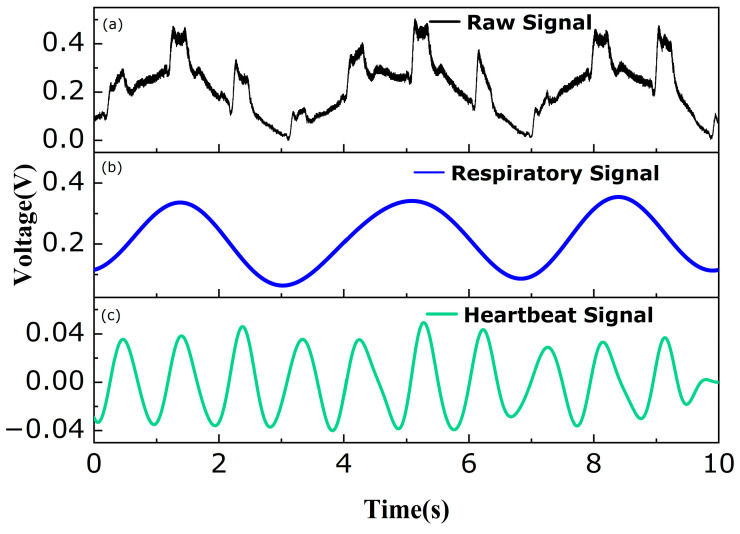
Demonstration of vital sign signal monitoring. (**a**) Raw signal. (**b**) Separated respiratory signal. (**c**) Separated heartbeat signal.

**Figure 8 sensors-24-02209-f008:**
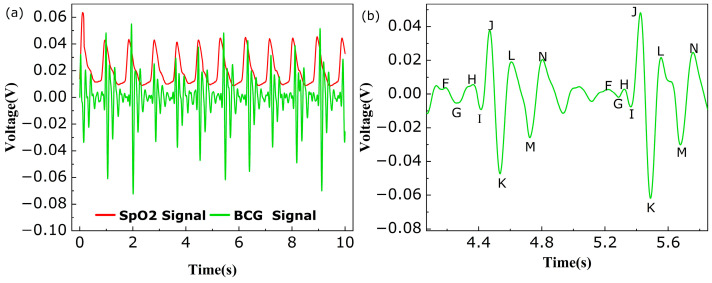
(**a**) Comparison between SpO2 signal and BCG signal; (**b**) a single BCG signal.

**Table 1 sensors-24-02209-t001:** Parameters of optical fiber.

Fiber Type	n_core_/n_clad_	Diameter (μm)
SMF	1.451/1.444	8.3/125
SIMMF	1.458/1.444	105/125
GIMMF	1.458 × sqrt (1 − 0.019 × (r/25)^2^)/1.444	50/125

**Table 2 sensors-24-02209-t002:** Information about experimental staffing.

Subject No.	Gender	Age	Weight (kg)	Height (cm)
1	Female	25	51	160
2	Female	24	60	158
3	Female	24	58	175
4	Female	25	50	165
5	Female	23	51	167
6	Male	23	62	160
7	Male	24	65	165
8	Male	24	55	158
9	Male	25	60	175
10	Male	24	70	185

**Table 3 sensors-24-02209-t003:** Error of HR and RR measured by the SHMF-MZI microbend sensor.

Index	Error of HR (Mean ± SD)			Error of RR (Mean ± SD)		
	ME (bpm)	MAE (bpm)	MAPE (%)	ME (bpm)	MAE (bpm)	MAPE (%)
Female1	−1.4 ± 5.2	4.1 ± 3.4	5.7 ± 4.9	0.5 ± 1.6	1.4 ± 0.9	7.2 ± 4.8
Female2	−1.9 ± 6.3	4.9 ± 4.4	6.0 ± 5.5	0.4 ± 1.8	1.3 ± 1.3	5.5 ± 4.5
Female3	−2.0 ± 6.1	4.7 ± 4.3	6.5 ± 6.6	−0.6 ± 2.4	1.7 ± 1.8	11.6 ± 14.1
Female4	−0.9 ± 6.1	4.8 ± 3.8	6.9 ± 5.7	0.7 ± 1.7	1.4 ± 1.2	5.8 ± 5.1
Female5	−1.0 ± 4.5	3.3 ± 3.3	4.2 ± 4.4	−0.5 ± 2.7	2.1 ± 1.7	15 ± 13.9
Male1	0.4 ± 2.9	2.0 ± 2.0	3.5 ± 3.4	0.7 ± 1.3	1.2 ± 0.9	5.3 ± 3.9
Male2	2.3 ± 5.3	4.4 ± 3.7	6.0 ± 5.3	0.3 ± 1.3	1.1 ± 0.7	5.0 ± 3.1
Male3	−1.7 ± 2.8	2.4 ± 2.3	3.3 ± 3.2	0.1 ± 1.1	0.8 ± 0.8	4.8 ± 4.4
Male4	2.1 ± 5.3	3.6 ± 4.4	3.9 ± 4.5	1.1 ± 1.7	1.6 ± 1.2	9.4 ± 7.5
Male5	1.2 ± 5.3	4.0 ± 3.7	4.5 ± 4.1	0.6 ± 2.3	1.7 ± 1.7	10.6 ± 12.5
Sit	0.8 ± 5.4	4.0 ± 3.7	4.9 ± 4.5	0.3 ± 1.8	1.3 ± 1.4	7.1 ± 10.2
Stand	−0.1 ± 5.7	4.0 ± 3.9	4.7 ± 4.6	0.6 ± 2.1	1.6 ± 1.4	8.9 ± 8.9
Lie	−1.0 ± 4.7	3.3 ± 3.5	5.2 ± 5.5	0.1 ± 1.8	1.4 ± 1.2	8.3 ± 8.4
Summarize	−0.1 ± 5.3	3.8 ± 3.7	4.9 ± 4.9	0.3 ± 1.9	1.4 ± 1.3	8.1 ± 9.2

**Table 4 sensors-24-02209-t004:** Comparison of the HR measurement of the SHMF-MZI microbend sensor with others.

Device	ME (bpm)	MAE (bpm)	MAPE (%)	Cost	Length (mm)	Technology	Ref.
SHMF-MZI	−0.1 ± 5.3	3.8 ± 3.7	4.9 ± 4.9	Low	22	Intensity	This work
FBG	0.8 ± 5.9	3.8 ± 4.6	0.6 ± 7.6	High	5	Wavelength	[20]
Apple Watch	−1.7 ± 10	5.0 ± 9.0	5.5 ± 9.4	High	-	PPG	[34]

**Table 5 sensors-24-02209-t005:** The value and absolute error of the HR calculated by the BCG and SpO2 signal for the male volunteer 3.

BCG (bpm)	SpO2 (bpm)	Absolute Error (bpm)	ME (bpm)	MAE (bpm)	MAPE (%)
63	62	1			
63	63	0			
61	61	0			
62	63	1			
62	64	2			
65	65	0			
64	67	3			
66	66	0			
68	69	1			
67	67	0			
65	66	1			
64	64	0			
68	70	2			
62	62	0			
65	65	0			
66	65	1			
66	65	1			
65	66	1			
67	69	2			
65	66	1			
(mean ± SD)			0.55 ± 1.07	0.85 ± 0.85	1.28 ± 1.27

**Table 6 sensors-24-02209-t006:** Pearson coefficient calculated for different subjects and different postures.

Index	*r_X_*_,*Y*_ of HR	*r_X,Y_* of RR
Female1	0.91	0.88
Female2	0.77	0.74
Female3	0.85	0.64
Female4	0.69	0.84
Female5	0.94	0.75
Male1	0.62	0.87
Male2	0.79	0.86
Male3	0.86	0.83
Male4	0.93	0.65
Male5	0.76	0.75
Sit	0.85	0.91
Stand	0.89	0.87
Lie	0.77	0.91
Summarize	0.92	0.90

## Data Availability

Data are contained within the article.

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
