# Peer review of "A Wearable Sandwich Heterostructure Multimode Fiber Optic Microbend Sensor for Vital Signal Monitoring"

_sensors, 2024, doi:10.3390/s24072209_

Round 1

Reviewer 1 Report

Comments and Suggestions for Authors

The authors proposed a highly sensitive sandwich heterostructure multimode optical fiber microbend sensor for vital signal monitoring. The optimal parameters of the sandwich heterostructure multimode fiber Mach- Zehnder interferometer (SHMF-MZI) were confirmed by the Rsoft software. The encapsulation of SHMF-MZI by the microbend deformer could improves its coupling efficiency, leading a very high press sensitivity of about 98dB/N. The proposed wearable sensor can greatly improve the response to HR signals. The monitoring performance of its HR, RR and BCG signals are as good as those of the medically approved commercial monitors. Therefore, I recommend this work to be accepted after addressing the following issues:

(1)    In this work, the authors have placed the wearable SHMF-MZI microbend sensor in a location near the heart. It would be beneficial to provide more explicit details about the precise positioning of the sensor. And what effect does the position of the sensor have on the monitoring effect? The author should explain as appropriate.
(2)    In this work, the respiratory signal was extracted from the original signal using an FFT filter, why the heartbeat signal uses a zero-phase filter? The authors should give a brief explanation.
(3)    In Table 3, why the HR error for female is higher than for male? The authors should give a reasonable explanation.
(4)    The lying position measurement yields the least amount of noise, thus indicating a higher level of accuracy compared to sitting and standing positions. However, why the lying MAPE and its corresponding SD exhibit the largest values. The authors should give a reasonable explanation.
(5)    There are some mistakes in the manuscript.    Please check carefully.    According to the data presented in Table 3, the maximum value of ME should be 2.3 bpm but not 2.5bpm in the text, and the maximum SD values for the ME and MAE groups should be interchanged for the error of HR in the text, and the same problem exists for the error of RR. The cov(X, Y) in equation (9) should be changed to Cov(X,Y), etc.
(6)    It is also suggested that the “Standard deviation” should use its abbreviation (SD) in Table 3.

(7)    Suggest the authors enhance the introdcution part with one latest literatures on this topic:10.1021/acsami.3c16165

  •  

Reviewer 2 Report

Comments and Suggestions for Authors

 This manuscript introduces a wearable optical fiber micro bend sensor, which combines a sandwich heterostructure multimode fiber Mach-Zehnder interferometer (SHMF-MZI) with a micro bend deformer for the simultaneous measurement of RR and HR. The manuscript is well-organized, and both the study's information and experimental methodology are intriguing. However, the authors could provide a more thorough explanation of their work. Overall, I believe this manuscript is suitable for publication in Sensor after significant revision.

  1. - The authors should explicitly mention that they conducted a detailed analysis and optimization of the parameters of the SHMF-MZI sensor and the micro bend deformer.

  2. - Stability and Reusability: It would be beneficial if the authors could elaborate on the sensor's stability and its potential for reuse.

  3. Conclusion: -While the conclusion appears satisfactory, it would be advisable for the authors to discuss the main limitations of their study.

  4. -Grammar Several grammatical errors, such as the continuous omission of 'a' or 'the' before specific words, detract from the clarity of the author's writing. The authors should thoroughly review the manuscript for such language issues.

Reviewer 3 Report

Comments and Suggestions for Authors

The paper, titled "Development of a Highly Sensitive Sandwich Heterostructure Multimode Optical Fiber Microbend Sensor for Vital Signal Monitoring," introduces a new design for such a sensor. One advantage of the article is its focus on a current research topic and its extensive reference list (34 references).

However, the reviewer has raised some questions:

  1. -How does the sensor handle changes in temperature during measurements, and how does this affect the output signal?

  2. - What is the impact of bending the fiber on the output signal?

Author Response

请参阅附件。

Round 2

Reviewer 2 Report

Comments and Suggestions for Authors

The authors have addressed in detail the concerns raised by the referees and the manuscript is suitable for publication.